# Drug-Induced Acute Pancreatitis in Adults: Focus on Antimicrobial and Antiviral Drugs, a Narrative Review

**DOI:** 10.3390/antibiotics12101495

**Published:** 2023-09-29

**Authors:** Angelo Del Gaudio, Carlo Covello, Federica Di Vincenzo, Sara Sofia De Lucia, Teresa Mezza, Alberto Nicoletti, Valentina Siciliano, Marcello Candelli, Antonio Gasbarrini, Enrico Celestino Nista

**Affiliations:** 1Center for Diagnosis and Treatment of Digestive Diseases, Gastroenterology Department, Fondazione Policlinico Universitario A. Gemelli, IRCCS, 00168 Rome, Italy; delgaudioangelo@gmail.com (A.D.G.); covellocarlo@gmail.com (C.C.); federica.divincenzo30@gmail.com (F.D.V.);; 2Pancreas Unit, Centro Malattie Apparato Digerente, Medicina Interna e Gastroenterologia, Università Cattolica del Sacro Cuore, Fondazione Policlinico Universitario A. Gemelli, IRCCS, 00168 Rome, Italy; teresa.mezza@policlinicogemelli.it (T.M.); alberto.nicoletti@guest.policlinicogemelli.it (A.N.); 3Laboratory and Infectious Diseases Sciences, Fondazione Policlinico Universitario A. Gemelli, IRCCS, 00168 Rome, Italy; valentina.siciliano@policlinicogemelli.it; 4Emergency, Anesthesiological and Reanimation Sciences, Fondazione Policlinico Universitario Agostino Gemelli—IRCCS, Università Cattolica del Sacro Cuore, 00168 Rome, Italy; marcello.candelli@policlinicogemelli.it

**Keywords:** acute pancreatitis, drug-induced acute pancreatitis, antimicrobial drugs

## Abstract

Acute pancreatitis (AP) is an acute inflammation of the pancreas caused by the activation of digestive enzymes in the pancreatic tissue. The main causes of AP are cholelithiasis and alcohol abuse; less commonly, it can be caused by drugs, with a prevalence of up to 5%. Causal associations between drugs and pancreatitis are largely based on case reports or case series with limited evidence. We reviewed the available data on drug-induced AP, focusing on antimicrobial drugs and antivirals, and discussed the current evidence in relation to the classification systems available in the literature. We found 51 suspected associations between antimicrobial and antiviral drugs and AP. The drugs with the most evidence of correlation are didanosine, protease inhibitors, and metronidazole. In addition, other drugs have been described in case reports demonstrating positive rechallenge. However, there are major differences between the various classifications available, where the same drug being assigned to different probability classes. It is likely that the presence in multiple case reports of an association between acute pancreatitis and a drug should serve as a basis for conducting prospective randomized controlled trials to improve the quality of the evidence.

## 1. Introduction

Acute pancreatitis (AP) is an acute inflammation of the pancreas resulting from intrapancreatic activation of digestive enzymes. This condition can lead to pancreatic necrosis, organ failure, and multiple organ dysfunction, with a mortality rate of 1–5% [1,2]. In addition, acute pancreatitis may be associated with significant short- and long-term morbidity, with recurrent symptoms and, in severe cases, exocrine and/or endocrine pancreatic failure [3]. The diagnosis of AP requires the identification of two of the following criteria: (1) abdominal pain suggestive of pancreatitis, (2) serum amylase or lipase levels at least three times the upper normal limit, and (3) imaging findings consistent with pancreatitis on computed tomography (CT) or magnetic resonance imaging (MRI) [1,4,5]. It is clear that the first two criteria alone are sufficient for diagnosis; thus, imaging can be performed later to better identify complications of acute pancreatitis. AP can be classified into mild, moderate, and severe forms according to the Atlanta criteria [1]. 

Mild acute pancreatitis is not associated with local or systemic complications and often resolves within the first week. Usually, mild AP does not result in organ failure. It is the most common form. Moderate AP is associated with local complications or exacerbation of concomitant diseases; transient organ failure may occur. In contrast, severe acute pancreatitis is defined by persistent organ failure (i.e., organ failure > 48 h). Local complications include pancreatic and peripancreatic necrosis (sterile or infected), peripancreatic fluid collections, pseudocysts, and parietal necrosis (sterile or infected) [1]. The main causes of AP are cholelithiasis and alcohol abuse, which account for more than 80% of cases [6,7]. Less commonly, it can also be caused by medications, with a prevalence ranging from 2 to 5.3% [3,7]. In this narrative review, we focused on the role of pharmacologic agents in causing acute pancreatitis, particularly antimicrobial and antiviral agents, considering the widespread use of these therapies and the absence of specific work on these types of drugs in the literature.

## 2. Drug-Induced Pancreatitis

According to the Agency for Drugs and Medical Devices Adverse Drug Event Reporting Database, drug-induced acute pancreatitis (DIAP) accounted for 0.17% of all adverse drug reactions (ADRs) [8]. Pharmacologic agents potentially responsible for acute pancreatitis include nearly 200 drugs [8,9]. However, causal relationships are largely based on case reports or case series with limited evidence. Consequently, an important indication of a causal relationship between a drug and AP is the recurrence of AP after reintroduction of the drug after discontinuation and recovery from a previous episode [10]. However, for ethical reasons, drugs in which rechallenge episodes are performed are quite rare. Another key concept in determining the association between a particular drug and an adverse event is latency; latency is the time between the assumption of the drug and the occurrence of AP. Several systematic reviews use latency categories (e.g., <24 h, 1–30 days, >30 days), and if >75% of case reports for a drug fell into the same latency category, this was considered good evidence of an association (adequate latency) [9,11,12]. Currently, only three drugs have been associated with the development of AP in a randomized controlled trial (6-mercaptopurine, azathioprine, and didanosine) [9,11,12,13]. 

In 2007, based on case reports in the literature, Badalov et al. classified drugs potentially responsible for the development of AP into probability classes based on case reports from the literature (Table 1) [14]. This classification was revised in 2019 by C.R. Simons-Linares et al. The first class includes drugs with a positive rechallenge. This class is further divided into two subclasses, a and b, depending on whether other possible causes were excluded or not. Class II includes drugs for which at least four case reports show similar latency. Class III includes drugs for which there are at least two case reports, but which do not meet the criteria of the above classes. The final class (IV) includes the drugs with the weakest evidence, for which only one case report was published [12]. This classification was updated by Wolfe et al. in 2020 [11]. 

However, there are some differences between these two classifications; methodologically, Wolfe et al. developed the rules of their classification a priori, before performing the literature search, whereas the system of C.R. Simons-Linares et al. is based on their review data. In addition, Wolfe et al. applied more stringent criteria in selecting case reports by including only cases of AP diagnosed according to currently accepted diagnostic criteria and excluding cases in which AP was associated with drug combinations (Table 2). In this classification, the classes differ from the previous one; class I was further divided into three subclasses (a, b, c). Class Ic includes drugs for which there is at least one case report with no positive rechallenge, excluding other causes. Drugs for which there are at least two case reports without evidence of rechallenge but with and without exclusion of other causes are classified as class II and class III, respectively. With adequate latency in class II and insufficient latency in class III [11]. 

More recently, Saini et al. have presented another classification based on an evidence-based approach that includes a comprehensive review of randomized controlled trials (RCTs), cohort studies, pharmacoepidemiologic analyses, and case reports and differs from previous classifications. Table 3 provides an overview of this classification system. They divided drugs considered to trigger AP into four groups according to the quality of the evidence. Thus, class I included RCTs (high evidence), whereas case–control studies and pharmacoepidemiologic studies were classified as class II. Classes III (divided into a, b, and c) and IV included case reports according to quality, presence of rechallenge, adequate latency, and number of reports [9].

### Main Mechanisms of Drug-Induced Pancreatitis

Most of the mechanisms underlying drug-induced pancreatic injury have not been fully elucidated or demonstrated. One of the most common is probably an idiosyncratic delayed immunologic or T-cell-mediated response rather than intrinsic toxicity [13,15]. This has been demonstrated for azathioprine, 6-MP, and sulfasalazine in both in vitro trials and population-based studies [10,16]. Idiosyncratic reactions do not appear to be directly related to dose, and symptoms differ from the pharmacologic effect of the drug. The proposed mechanism for idiosyncratic reactions to drugs is not certain. However, it may be a reactive metabolite of the drug that binds to specific proteins and causes an immune system response (haptenes hypothesis). This response could be triggered by injury or cellular stress (hazard hypothesis) and stimulate the activation of an immune response by the co-stimulation of T lymphocytes [15].

Although the exact mechanism of AP, triggered by thiopurines and antibiotics, remains unknown, there is increasing evidence that certain human leukocyte antigen (HLA) alleles may be predisposed to this type of hypersensitive reaction [17].

Other proposed mechanisms include the hypothesis of a toxic effect on the pancreatic cell membrane (as in the case of valproate) or the formation of edema in the pancreatic ducts, which may occur with angiotensin-converting enzyme inhibitors via the kallikrein/kinin pathway [18]. In addition, some drug therapies have been associated with the occurrence of hypertriglyceridemia pancreatitis [19]. According to some studies, most antiviral protease inhibitors (PI) are associated with a significant increase in plasma triglyceride concentrations; in particular, hypertriglyceridemia is observed more frequently after combination therapy with ritonavir or lopinavir/ritonavir than with other PI -based combinations. As described below, ritonavir and lopinavir/ritonavir-based regimens have been associated with an increased risk of hyperlipidemic pancreatitis [19,20].

Another possible causative mechanism is mitochondrial toxicity [19,21]. This appears to be associated with the use of antiviral drugs such as nuclear reverse transcriptase inhibitors to treat human immunodeficiency virus (HIV) infections [10,22]. This toxicity appears to be due to the inhibition of mitochondrial DNA replication by suppression of the activity of key enzymes involved in mitochondrial DNA replication. Although acute pancreatitis is a well-described complication of HIV infection, an increased incidence of this condition has been reported after the introduction of reverse transcriptase inhibitors in HIV patients [21,23].

Considering antibiotics, it is postulated that tetracyclines may cause DIAP either through a toxin-mediated action of an unknown metabolite or a direct toxic effect on the pancreas due to the supratherapeutic bile concentration of tetracyclines [18]. Pharmacological experiments with tetracyclines have shown that the biliary concentration of minocycline is ten times higher than the serum concentration, and similar results have been observed with tigecycline [24,25].

Another speculative mechanism of metronidazole-induced pancreatitis is the production of redox cycling and hydrogen peroxide, superoxide, and other free radicals under aerobic conditions. These redox-active compounds are toxic to pancreatic β-cells. Oxygen free radicals have been associated with the initiation of pancreatitis [26]. Finally, in the case of opiates, the proposed mechanism is a stimulation of μ-receptors causing hypercontraction of the Oddi sphincter that increases the basal pressure and amplitude of contraction and induces a reflux of pancreatic enzymes into the pancreatic duct [27]. Figure 1 summarizes these mechanisms. 

**DIAP: Drug**-**induced acute pancreatitis**

The etiopathogenetic mechanism most involved in the development of DIAP from antimicrobials and antivirals appears to be the delayed idiosyncratic or lymphocyte-T-mediated mechanism. Pancreatic damage caused by the immune system also seems to be associated with certain human leukocyte antigen (HLA) alleles which may be predisposed to this type of hypersensitive reaction. Other pathogenetic mechanisms have been postulated for these drug classes. For example, hypertriglyceridemia induced by protease inhibitor antivirals may increase the risk of developing DIAP. Furthermore, mitochondrial toxicity, oxidative damage, and biliary supersaturation by metabolites or the drug itself have been suggested as additional mechanisms involved in DIAP by reverse transcriptase inhibitors, metronidazole, and tetracyclines, respectively.

## 3. Antibiotics

In this chapter, we have divided antibiotics according to the most accepted pharmacological classification, as follows: inhibitors of cell wall synthesis, inhibitors of protein synthesis, inhibitors of cell metabolism, inhibitors of nucleic acid synthesis, and antibiotics that alter membrane permeability [28]. Table 4 summarizes the antibiotics associated with acute pancreatitis.

### 3.1. Nucleic Acid Synthesis Inhibitors

#### 3.1.1. Metronidazole and 5-Nitroinidazoles

Metronidazole is a bactericidal antibiotic that attacks host microorganisms, causes severe DNA damage, and inhibits the synthesis of nucleic acids. It is the drug of choice for treating infections caused by anaerobic bacteria, *Helicobacter pylori*, and protozoan infections (e.g., giardiasis).

Metronidazole has been mentioned in several case reports on PA, with three patients having a positive rechallenge and all having a favorable outcome [29]. Based on these data, it is classified as class Ia according to both the classification of Roberto Simons-Linares et al. and the classification of Diana Wolfe et al. We would like to point out that three other cases have been described in the literature that are not included in this narrative review, two of them with a positive rechallenge [11,12,30].

Andrei Barbulescu et al. conducted a large population-based case–control study of 5996 patients with AP in Sweden between January 2006 and December 2008. The odds ratios of metronidazole exposure in individuals with AP were 4 and 11 for single oral exposure and for combined regimens for treatment of *H. pylori*, respectively [31]. In addition, Nørgaard M. et al. reported an eightfold increased risk of acute pancreatitis in individuals exposed to metronidazole when it was combined with other drugs used to treat *H. pylori* [32]. As a result of these two case–control studies, metronidazole was placed in class II of the Saini et al. classification [9].

Secnidazole and tinidazole are 5-nitroimidazole derivatives with similar properties to metronidazole, except for a longer-lasting blood concentration, used for hepatic amibiasis, giardiasis, and bacterial vaginosis [33].

Only one case of AP has been described after the use of secnidazole and another case after tinidazole ingestion [34,35]. All other underlying diseases or medications associated with AP were ruled out; however, in both cases, the rechallenge was negative. Therefore, the drugs met the criteria for class IV in the reviewed classification of C. Roberto Simons-Linares et al. At the same time, they belong to class Ic according to the classification system of Diana Wolfe et al. However, of the two drugs described, only secnidazole was included in class IV in the review by Saini et al. [9,11,12].

#### 3.1.2. Fluoroquinolones

Fluoroquinolones are bactericidal antibiotics that inhibit DNA gyrase and topoisomerase IV, which are involved in DNA synthesis. They are broad-spectrum antimicrobial antibiotics with excellent efficacy against several Gram-negative (including *Pseudomonas* spp.), and some Gram-positive, and also atypical bacteria (e.g., Chlamydia, Legionella, Mycoplasma), and tuberculous and non-tuberculous mycobacteria. Several cases of acute pancreatitis triggered by fluoroquinolones have been described in the literature, but no pathogenetic mechanism has been postulated for this pharmacological class [36,37].

With respect to ciprofloxacin, two cases were described in the literature, with no consistent latency between cases and no rechallenge [36]. For these reasons, it was placed in class III of the classification by Roberto Simons-Linares et al. [12].

However, based on the data reported in the work of Sung HY et al., this drug was classified as class Ib by Wolfe et al. because one case documented a positive rechallenge without exclusion of other causes of AP [37]. However, it was not mentioned in the review by Saini et al. because the case reports cited were probably considered to be of low quality.

Levofloxacin was not included in the classifications of Simons-Linares, Wolfe, and Saini [9,11,12]. However, four cases of AP caused by levofloxacin have been described, including one in which rechallenge positivity was observed without definite exclusion of other causes of AP. However, all cases were characterized by a good outcome [38]. It should be noted that the aforementioned case of rechallenge positivity is marred by a lack of exhaustive details. We can conclude that levofloxacin, regardless of the risk class, should be considered as a drug that, probably, can induce DIAP.

Ofloxacin is not included in Roberto Simons-Linares and Wolfe’s classifications, although a case of DIAP triggered by the drug in question has been reported [39,40]. We can assume that this case was excluded because of the association of this drug with ornidazole. Therefore, it was classified as an orfloxacin–ornidazole association in class IV of the Saini classification because of uncertain rechallenge positivity.

For norfloxacin, only one case without rechallenge was described in the medical literature; therefore, it was placed in class IV by Wolfe et al. [11]. Norfloxacin was not included in the Simons-Linares et al. reviewed classification because the criteria used in the review did not include case reports in languages other than English. Instead, probably for reasons related to the low quality of the case report, it was not included in the classification of Saini et al. [9].

#### 3.1.3. Nitrofurantoin

Nitrofurantoin is an antibacterial drug of the nitrofuran class used mainly against urinary tract infections. It inhibits DNA and RNA functions by a mechanism that has not been fully elucidated, although nitro group bioreduction is thought to be an essential component of the mechanism of action.

The first case of DIAP associated with nitrofurantoin ingestion was described by Neils et al. in 1983 [41]. This case, in which rechallenge positivity was documented without clearly excluding other causes of pancreatitis, allowed to include nitrofurantoin in class Ib according to the classification of Roberto Simons-Linares et al.

Wolfe et al., on the other hand, assigned nitrofurantoin to class Ia because they included two other cases in the review, one of which was rechallenge-positive and other causes of AP were excluded with certainty [42,43]. Saini et al. agreed with the latter classification and therefore assigned it to class IIIb. Finally, all cases described in the literature indicated mild pancreatitis with a good outcome.

#### 3.1.4. Rifampicin

Rifampicin is a bactericidal antibiotic that belongs to class of rifamycins and exerts its mechanism of action by selectively inhibiting bacterial RNA synthesis and blocking transcription. Rifampicin is mainly used for the treatment of nontuberculous mycobacteriosis and tuberculosis and is characterized by a wide spectrum of adverse effects and drug–drug interactions, which, however, are not the subject of this review. In this context, only one case of DIAP caused by rifampicin with a positive rechallenge and uncertain exclusion of other causes of pancreatitis has been described [11].Therefore, it was placed in class Ib of the Wolfe classification and excluded by Simons-Linares et al. and Saini et al. because the reported case was not available in English [11,12].

### 3.2. Protein Synthesis Inhibitors

#### 3.2.1. Tetracyclines

Tetracyclines are bacteriostatic antibiotics that inhibit protein synthesis by binding to the 30 s subunit of the bacterial ribosome. At least one case of DIAP has been described for drugs in this group (tetracycline, doxycycline, minocycline, tigecycline, and dememocycline [44,45,46].

Tetracycline is not included in the classification of Roberto Simons-Linares et al. because it does not meet the inclusion criteria used, which is probably related to the lack of availability of relevant details in the case.

However, we found three case reports that allowed its classification as class Ia by Wolfe et al. and class IIIb by Saini et al. [9]. Of the reported cases, one is characterized by a positive rechallenge and definitive exclusion of other causes of pancreatitis [24,45,47,48]. 

Considering Doxycycline, eleven cases of doxycycline-induced acute pancreatitis were reported, four of which were not included in the Roberto Simons-Linares and Wolfe classification. However, because of the different inclusion criteria in the classifications considered, doxycycline was placed in different risk classes. In three of the cases mentioned above, there was no consistent latency between cases and no rechallenge and thus this drug was included in class III in the Simons-Linares classification. In contrast, Wolfe et al. included four different cases from the above-mentioned review, all of which had no positive rechallenge and in which other causes of pancreatitis have been excluded; therefore, the drug was classified as Ic [11,44,46,49,50,51,52,53,54,55].

Saini et al. considered four cases reported in the literature, two of which were not included in the above classifications, all of which had a consistent latency period with no positive rechallenge [56]. The drug was therefore classified as IIIc [9].

Mild to moderate forms of pancreatitis were described in all reported cases, with the exception of a single case of severe pancreatitis.

With respect to tigecycline, ten cases of associated DIAP have been described [25,57,58,59,60,61]. However, because of the variable quality of reported cases, this drug is placed in different, often divergent, classes for the classifications considered. Wolfe et al. and Simons-Linares considered it as a high-risk drug for DIAP, placing it in classes Ia and Ib, respectively. This class difference, consistent with the above criteria, was observed because Wolfe et al. included a clinical case describing rechallenge positivity and exclusion of other pancreatitis etiologies that was not evaluated by Simons-Linares [11,25,62,63,64]. In contrast, Saini et al. included only two cases, none of which had rechallenge positivity and/or consistent latency and thus they were included in class IV [9,65]. Overall, mild to moderate pancreatitis with a good outcome occurred in all reported cases.

Minocycline and demeclocycline were not included in the classification of Saini et al. For minocycline, three case reports were described, and none of these reports described a positive rechallenge [66,67,68]. Due to a disagreement regarding the exclusion of other etiologies of pancreatitis and the concordance of latency among the cases considered, minocycline was included in classes III and Ic of the Simons-Linares et al. and Wolfe classifications, respectively.

In contrast, only one case of associated DIAP is suspected for demeclocycline. A positive rechallenge and complete exclusion of other causes of pancreatitis were not reported and thus it was placed in class IV by Wolfe et al. [11]. 

#### 3.2.2. Macrolides

Macrolides are bacteriostatic antibiotics that inhibit bacterial protein synthesis by binding to the 50 S ribosomal subunit. The spectrum of activity of this class of drugs includes Gram-positive and Gram-negative cocci and atypical bacteria [69].

Several cases of DIAP have been described for several drugs of this class, such as erythromycin, clarithromycin, and roxithromycin.

Twelve cases of erythromycin-induced AP have been described in the literature, at least one of which had a positive rechallenge [70,71,72,73,74,75,76,77,78,79,80,81]. However, in the classifications considered, there is a wide variability in the inclusion of the different drug classes, due to the poor quality of the details reported in the cases, especially regarding the definitive exclusion of other causes of pancreatitis. All cases reported in the literature were mild forms of pancreatitis with a good outcome.

It was classified as class Ia in the Wolfe et al. classification but it was assigned to class III and IV by Simons-Linares et al. and Sai [9,12].

Six cases of DIAP associated with clarithromycin have been reported in the literature, with no one positive rechallenge [82,82,83,84,85,86] For the same reasons explained earlier for erythromycin, there is considerable discordance in the classification systems used for clarithromycin. While it was classified as class Ic in the Wolfe et al. classification, C. Roberto Simons-Linares placed it in class III, and Siani et al. excluded this drug from their review.

It is worth mentioning two cases of mild pancreatitis induced by roxithromycin, one of which was due to the combination with betamethasone [86]. In both cases, no rechallenge positivity was described, and the drug was included in class IV of all classifications considered in this review [9,11,12].

### 3.3. Trimethoprim/Sulfamethoxazole—Inhibitors of Cellular Metabolism

Sulfonamides such as sulfamethoxazole and trimethoprim interfere with folic acid biosynthesis by competitive inhibition of dihydropteroate synthetase and dihydrofolate reductase, respectively. These two drugs are often combined and are the treatment of choice for nocardiosis, pneumocystosis, Whipple’s disease, and *Stenotrophomonas* sp. An association between trimethoprim/sulfamethoxazole and DIAP was established in eight case reports [87]. Two case reports described a positive rechallenge without other possible causes of pancreatitis, and the other two cases resulted in death [88,89,90,91]. One of the case reports described a positive lymphocyte stimulation test with sulfamethoxazole, which may suggest that sulfamethoxazole is the main culprit of the reported DIAP [92,93].

According to the classifications of Wolfe and Simons-Linares et al., trimethoprim/sulfamethoxazole was classified as a high-risk drug in class Ia [12]. According to the criteria used by Saini et al., the drug was classified as class IIIb due to the positive rechallenge in the absence of consistent latency among cases [9].

### 3.4. Inhibitors of Cell Wall Synthesis

#### 3.4.1. Beta-Lactams

Penicillins belong to the group of beta-lactam antibiotics that bind to bacterial transpeptidases and inhibit the synthesis of peptide crosslinks between peptidoglycan chains. Ampicillin and amoxicillin/clavulanate are the two penicillins most likely to be associated with AP [94].

Hanline MHJ has described a case of AP caused by ampicillin in which rechallenge positivity was observed without completely ruling out other causes of pancreatitis [31]. Therefore, the drug was classified as Ib by Wolfe et al. However, Simons-Linares et al. questioned the positive rechallenge because of insufficient data reported by the authors and reclassified ampicillin as III.

For amoxicillin/clavulanate, only two cases of associated DIAP were reported in the literature [95,96]. In neither case was rechallenge positivity described, and in one of these cases, all other causes of pancreatitis were not excluded. However, because of the different inclusion criteria in the systematic reviews, amoxicillin/clavulanate was classified as class Ic by Wolfe and as class IV by Simons-Linares et al. Finally, no penicillins were included in the review by Saini et al. [9,11,12].

Ceftriaxone is the only cephalosporin for which a possible association with DIAP has been seen; it has excellent activity against most Gram-negative and some Gram-positive bacilli [97].

At least two cases of DIAP caused by ceftriaxone have been described, with no consistent latency between cases and no positive rechallenge [97,98,99,100,101]. Therefore, it was included in classes II and III by Wolfe et al. and Simons-Linares et al., respectively. Saini et al. did not include it in their review [9,11,12]. 

#### 3.4.2. Isoniazid

Isoniazid is an antitubercular antibiotic that inhibits cell wall synthesis and acts specifically on the synthesis of mycolic acids, which are essential components of the cell wall of the bacterium Mycobacterium tuberculosis [102].

Isoniazid has been associated with pancreatitis in several patients, either during chemoprophylaxis or the treatment of tuberculosis [102,103,104,105,106,107,108,109,110,111,112,113,114]. A positive rechallenge was described in a total of seven patients, with other causes excluded in seven patients. Mild to moderate pancreatitis with a good outcome occurred in all reported cases.

Consistent with the Wolfe and Simons-Linares classifications, isoniazid was considered as a high-risk class Ia drug [11,12]. Similarly, Saini et al. classified it as class IIIb because positive rechallenge was confirmed in the absence of adequate latency among cases.

## 4. Antivirals

In the description of antiviral agents associated with DIAP, they are divided according to the most widely accepted pharmacological classification into protease inhibitors, reverse transcriptase inhibitors, DNA polymerase inhibitors, and individual drugs that do not belong to these classes. Table 5 summarizes the antivirals associated with acute pancreatitis.

### 4.1. Protease Inhibitors 

AP has been described in patients treated with protease inhibitors (PIs) [20]. As reported, most evidence of the causal relationship between AP and PIs comes from case reports. However, a pharmacoepidemiologic study based on analysis of the Food and Drug Administration’s Adverse Event Reporting System (FAERS) database was published in 2021 [20]. In this study, 12 of the 15 PIs studied were associated with PA-related adverse events. Consistent with this, Saini et al. classified the entire drug class as II. Nevertheless, some drugs in this class deserve separate consideration, such as ritonavir, an antiretroviral drug that is part of Highly Active Anti-Retroviral Therapy (HAART) for HIV infection. The main reason for its use in this therapy is its ability to inhibit the same enzyme that metabolizes other protease inhibitors. Qin et al. reported numerous cases of AP during ritonavir therapy in the above-mentioned study, with the majority of reports coming from the ritonavir/dasabuvir/ombitasvir/paritaprevir combination (54 and 64 reports, respectively) [20]. This combination appears to have the earliest onset of AP among all PI regimens studied, and ritonavir-based PI regimens may trigger AP earlier than other PI regimens. Ritonavir and lopinavir/ritonavir appeared to be associated with a higher risk of death. In contrast, based on case reports, ritonavir is classified as III by Simons-Linares et al. and IV by Wolfe et al. [11,12]. In addition, using proportionality analysis and Bayesian analysis, Quin et al. found that indinavir appears to have the strongest association with AP among all PI regimens. Indinavir is another antiviral protease inhibitor [20]. However, it is not included in the Wolfe and Simons-Linares classification system, probably due to the insufficient number of case reports and the frequent use of this drug in combination with other drugs.

Nelfinavir, a drug used in the treatment of (HIV-1) infection, recently proposed in the therapy of advanced pancreatic carcinoma, has presented five reports of acute pancreatitis according to Quin et al., with fatal events occurring in one patient [20,115,116,117]. In addition, one case of severe AP during the use of nelfinavir has been reported in the literature [118]. Nelfinavir is classified as group Ib according to both the classifications of Simons-Linares et al. and Wolfe et al. [11,12]. However, other drugs were also present in the reported case, but only showed a positive rechallenge. 

Telaprevir is a direct-acting antiviral drug, a selective protease inhibitor, used to treat HCV infection [119]. It was not included in the study by Quin et al. because it is not used to treat HIV such as the other drugs studied.

Based on the case report, it is classified as group Ia according to Simons-Linares et al. and Wolfe et al., due to both positive rechallenge and exclusion of all other causes of AP [119]. However, only cases of mild pancreatitis are reported in the literature [10].

According to individual case reports, other drugs with a less obvious association with AP are atazanavir, darunavir, and boceprevir [20,120]. The first two were not included in the Wolfe and Simons-Linares classification, but pancreatitis is listed in the product information as a risk factor and/or adverse drug reaction (ADR) [10]. In addition, according to Quin et al., 52 cases with 6 deaths were reported for atazanavir, while 26 cases with 2 deaths were reported for darunavir.

Instead, boceprevir was classified as class IV according to Simons-Linares and class 1c according to Wolfe et al. because the published case reports excluded the other causes of AP [11,12,121].

### 4.2. Interferon α2b/Ribavirin 

These drugs are used in combination in the treatment of HCV infection. Interferon a2b exerts antiproliferative and immunomodulatory effects through a partially unknown mechanism of action, whereas ribavirin, a nucleoside analogue (guanosine analogue with modified nitrogen), acts by inhibiting nucleoside synthesis. This combination has been associated with the occurrence of AP [121]. According to Simons-Linares et al., it is included in class III. Saini et al. included this association in class IIIc, by applying less stringent exclusion criteria and for the adequate latency between cases.

Wolfe et al. did not include this combination in their hypothesis because of the drug association. Looking at the individual drugs, ribavirin alone is not mentioned in this classification, whereas interferon alpha is included in class Ia [11].

In terms of severity, pancreatitis in reported cases was often severe enough to require hospitalization, although symptoms resolved rapidly after discontinuation of antiviral therapy [121].

### 4.3. Remdesivir

Remdesivir is an antiviral drug of the nucleotide analogue class, an RNA polymerase inhibitor, originally used to treat the Ebola virus; it was widely used during the COVID-19 pandemic and has been shown to reduce recovery time and patient mortality. Since then, several case reports and an observational study of the occurrence of AP in patients receiving this therapy have been published [122]. In 2021, it was published in the WHO Pharmaceutical Bulletin that administration of remdesivir appears to be associated with AP [122].

A retrospective observational study of 201 patients with SARS-CoV-2 infection treated with remdesevir showed a significant increase in pancreatic enzyme levels and/or acute pancreatitis in 23 cases. Six patients had clinical pancreatitis, and three also had radiographic evidence of AP [123].

However, in this study and in many other reports examined, patients were also treated with other potentially responsible drugs, such as steroids. In addition, the occurrence of AP may also be caused the infection itself, making etiologic differentiation difficult [122,123,124]. Interestingly, a recent retrospective study examined the relationship between COVID-19 and pancreatitis in 48,012 patients hospitalized in New York, of whom 11,883 were COVID-positive. A total of 189 patients had adequate diagnostic criteria for pancreatitis at the time of admission, of whom 32 were COVID-19-positive. Among COVID-19 patients, the most frequent aetiology was idiopathic (69%), compared to 21% of COVID-19-negative patients (*p* < 0.0001). These data reveal the causative role of SARS-CoV-2 infection in acute pancreatitis [125].

In contrast to these data, AP has not been reported in other studies conducted in more extensive case series (538 or 384) of patients receiving remdesivir [126].

### 4.4. Reverse Transcriptase Inhibitors

This class includes drugs that can inhibit the viral replication process by blocking the transcription of viral RNA into pro-viral DNA; these drugs are further divided into nucleoside inhibitors, nucleotide inhibitors, and non-nucleoside inhibitors depending on the mechanism of enzyme inhibition [127].

Within this broad class, some drugs have been associated with the risk of inducing AP with varying degrees of evidence, including the following:

#### 4.4.1. Didanosine

Dinanosine is a nucleoside analogue that is active against HIV and is used in combination with other retroviral drugs as part of highly active antiretroviral therapy (HAART). Specifically, it is a synthetic analogue of deoxyadenosine; it consists of ten carbon atoms, twelve hydrogen atoms, four nitrogen atoms and three oxygen atoms (C_10_H_12_N_4_O_3_); it is administered orally and metabolized renally; and it has low bioavailability because it is hydrolyzed at an acidic pH [128].

Together with azathioprine, they are the drugs most commonly associated with the development of AP. They are supported by several randomized controlled trials and prospective studies conducted mainly in the 1990s [126,127,128].

For these reasons, it was classified as class I according to Saini et al. [9].

Considering the case reports, it was classified as class I according to Wolfe and class II according to Simons-Linares et al. [11,12]. In addition, among the listed reports, there are also cases of didanosine-mediated severe acute pancreatitis and death [129,130,131,132,133].

Most studies agree that the risk of AP increases in a dose-dependent manner with plasma levels of the drug [131,132]. 

Interestingly, didanosine has also been associated with the risk of chronic pancreatitis, with a high cumulative dose risk [131]. 

There is a scarcity within the literature on the possible mechanism responsible for pancreatic toxicity, but it seems to be responsible for mitochondrial damage [131].

#### 4.4.2. Lamivudine 

Lamividine is a potent nucleoside analogue that inhibits both type 1 and type 2 reverse transcriptase virus replication. Lamivudine is commonly used to treat HIV infection and hepatitis B [134]. Simons-Linares et al. classified lamivudine in class IV. However, there are reports in the literature characterized by a positive rechallenge without completely ruling out other causes of pancreatitis, particularly the presence of other drugs; therefore, it was later classified as class Ib by Wolfe et al. [11,12]. Clinically, no cases of severe pancreatitis have been reported [134].

#### 4.4.3. Less Associated Reverse Transcriptase Inhibitors

Other drugs belonging to this class of antivirals, but with weaker association, are abacavir, zidovudine entecavir tenofovir efavirenz, and nevirapine [10]. 

### 4.5. DNA Polymerase Inhibitors

DNA polymerase inhibitors are antiviral drugs used mainly to treat infections caused by herpes viruses (herpes simplex, varicella zoster, cytomegalovirus), but also for the treatment of hepatitis C and B [135,136]. Some agents in this class have been associated with the risk of developing AP; in particular, adefovir and famciclovir are the only ones included in the classifications discussed [135,136]. Adefovir, which is primarily used to treat hepatitis B, was classified as class Ic according to Wolfe et al. because a case of severe AP associated with adefovir use in a liver transplant recipient was reported in 2008 [11]. In that report, other causes of acute pancreatitis were ruled out; moreover, with regard to the mechanism responsible, the authors suspected direct mitochondrial toxicity from the drug, favored by ‘impaired renal and hepatic functions‘. However, both Simons-Linares and Saini place adefovir in the lowest level of evidence (class IV) because, in their opinion, other causes of pancreatitis have not been ruled out and it is a single reported case [9].

As for famciclovir, one case of severe acute necrotic hemorrhagic pancreatitis was reported in 1995 without exclusion of other possible causes; consequently, it was placed in class IV according to the Simons-Linares and Wolfe classification. Otherwise, acyclovir, ganciclovir, valganciclovir, cidofovir, and foscarnet are not included in the above classifications because there are no published case reports in the literature. However, the product information lists pancreatitis as a possible side effect [10].

## 5. Antifungals and Antiparasitics 

### 5.1. Pentamidine

Pentamidine isethionate is a diamino aromatic compound active against Pneumocystis and several protozoan pathogens, including *Leishmanii* spp., *T. b. gambiense*, *Trypanosoma*, *Pneumocistis jirovecii* [137].

Since the 1980s, numerous case reports have associated pentamidine in all its forms, intravenous, intramuscular, and aerosol, with the onset of AP, in some cases even with fulminant course [138,139,140,141,142,143,144,145,146,147,148,149,150,151,152,153,154,155,156,157]. Three patients in two case reports had positive rechallenge, although details to exclude other causes are not available. 

According to the drug classification system reviewed in 2019 by Simons-Linares et al.’s and also Wolfe’s classification, pentamidine was assigned to class Ib [11,12].

Saini et al. placed pentamidine in class IV, likely due to the different inclusion criteria of the case reports used [9].

### 5.2. Meglumine Antimoniate

Meglumine antimoniate is a pentavalent antimonials recommended as the first-line agent for the treatment of visceral, mucocutaneous, and cutaneous leishmaniasis [158]. The mechanisms of action and toxicity of pentavalent antimonials, including genotoxic effects, remain unclear, but may involve the inhibition of parasite glycolytic and fatty acid oxidative activity, which promotes oxidative stress-induced DNA damage [158,159].

In 1994, it was reported that a series of 49 patients had leishmaniasis, 48 (98%) of whom had chemical pancreatitis (hyperamylasemia and/or hyperlipasemia), although only half of these patients were symptomatic [160,161,162]. Since the 1990s, five case reports have described the occurrence of AP after the administration of meglumine antimoniate, one involving an infant and one involving a patient with the HIV infection and hepatic dysfunction [161,162,163]. Of the cases described, only one had a positive rechallenge. Therefore, it was assigned to class II in the classification by Simons-Linares et al. Alternatively, Wolfe et al. classified meglumine antimoniate as Ib in 2020 [11,12]. However, it does not fit into the more recent classification of Saini et al. for the same reasons given in the previous chapter for pentamidine.

### 5.3. Paromomycin

Paromomycin is an antimicrobial agent used to treat various parasitic infections, including amoebiasis, cryptosporidiosis, giardiasis, leishmaniasis, and tapeworm infestation [163]. Its mechanism of action is still unclear.

In 1995, Winston W. Tan et al. reported the case of a 39-year-old man with AIDS, who developed AP after receiving paromomycin for intestinal cryptosporidiosis [164]. A positive rechallenge was described when paromomycin was re-administered. However, because of the concurrent HIV and cryptosporidial infection, prior exposure to didanosine/octreotide/pentamidine, and history of alcoholism, the association was weak [164]. According to the reviewed classification system of Simons-Linares et al., it belonged to class IV, whereas Wolfe et al. classified paromomycin as class Ib due to the presence of a positive rechallenge [11]. In contrast, paromomycin, similarly to pentamidine and meglumine antimonate, does not fit the criteria to be included into the more recent classification by Saini et al. [9].

### 5.4. Stibogluconate

Sodium stibogluconate is a drug used to treat cutaneous, visceral, and mucosal forms of leishmaniasis [165,166]. Its mechanism of action involves a reduction in ATP and GTP synthesis, which contributes to the decrease in macromolecular synthesis and viability of leishmania.

A total of eighteen case reports describes DIAP in patients receiving sodium stibogluconate for the treatment of leishmaniasis [163,163,167,168,169,170,171,172,173,174,175]. In all reported cases, symptoms occurred within one month of starting treatment. A positive rechallenge was described in seven cases, while in eight cases there was an underlying disease that could be associated with pancreatitis, and in four cases the patients were also receiving other drugs that may be associated with DIAP [175]. Therefore, stibogluconate was classified as class Ib according to the drug classification system for assessing association with DIAP by Simons-Linares et al. and also by Wolfe et al. [11,12]. In contrast, similarly to pentamidine and meglumine antimonate, stibogluconate does not fit into the more recent classification review of Saini et al. [9]. 

### 5.5. Triazoles 

Triazoles are broad-spectrum antifungal agents that act by inhibiting ergosterol synthesis, the major membrane sterol of fungi. Clinically, they are mainly used to treat all types of invasive fungal infections. At least one case of DIP has been described for some drugs of this group (voriconazole and itraconazole). The specific mechanism of action of triazole-induced pancreatitis is still unclear. It is possible that the difference in distribution in pancreatic tissue explains why voriconazole or itraconazole are more often not tolerated by patients in comparison to fluconazole, which is better tolerated. Therefore, it is conceivable that plasma concentrations of the drug are higher than normal in patients who develop DIAP [175,176]. This could occur as a result of drug–drug interactions, impaired liver function, or polymorphisms of cytochromes such as CYP2C19, CYP2C9, and CYP3A4 that metabolize triazoles [9,176].

#### 5.5.1. Itraconazole

Regarding itraconazole, Passier et al. reported four cases of DIAP that developed after oral intake of itraconazole, two of which probably had a positive rechallenge, although this could not be demonstrated by laboratory or radiological investigations in the first episode. Other underlying diseases associated with AP were probably excluded. However, in 50% of cases, patients were taking other drugs concomitantly, which could not be excluded as a cause of DIAP [177]. Thus, itraconazole met the criteria for drug class Ic in the classification of Diana Wolfe et al. and for class IV in the classification of Siani et al., whereas it was excluded in the previous classification system [9,11].

#### 5.5.2. Voriconazole

Regarding voriconazole, Philip A. reported the case of a 16-year-old girl who developed AP during therapy with voriconazole for the treatment of invasive aspergillosis, with a positive rechallenge and exclusion of other causes of pancreatitis [163,178,179]. Thus, while voriconazole was not classified by Wolfe et al. and Siani et al., it met the criteria for drug class Ib in the classification of Diana Wolfe et al. In 2022, Song Qun Li et al. described a second case of acute DIAP, which occurred after taking voriconazole for more than a month in a 16-year-old girl treated for cryptococcal neoformans meningoencephalitis. All other underlying diseases or medications associated with pancreatitis were excluded and rechallenge was not described [179].

### 5.6. Artesunate

Artesunate is an artemisinin derivative antimalarial whose intravenous form is preferred to quinine for severe malaria [180]. Artesunate, like any other artemisinin, inhibits the polymerization of heme, generates ROS, destabilizes the parasite membrane, alkylates proteins, and inhibits PfATP6 [181]. Mahdi AS reported the case of a 34-year-old man with severe malaria who developed AP after 8 days of intravenous administration of artesunate [174,175,176,177,178,179,180,181,182,183,184]. Other causes of AP were ruled out; however, no rechallenge was performed. Therefore, the drug met the criteria for class IV in the classification of Simons-Linares et al., whereas it belongs to class Ic according to the classification system of Wolfe et al. It was excluded from the review of Siani et al. [9,12].

Table 6 summarizes the antifungals and antiparasitics associated with acute pancreatitis.

## 6. Materials and Methods

In this narrative review, we collected all the original reports that described the new onset of acute pancreatitis after the intake of antibiotics, antivirals, antifungals, and antiparasitics. 

A literature search was performed using the following electronic databases: PubMed, Scopus, and Embase databases for English and Spanish literature. The last search was run on 31 August 2023. The terms “drug-induced”, OR “medication-induced”, OR “drug-associated”, OR “medication-associated”, OR “chemically induced”, OR “drug” OR “medications” were matched with the word “pancreatitis”. All the terms were searched both as keywords and Medical Subject Headings (MeSH). We hand-searched the bibliographies of relevant (according to titles and abstracts) articles to provide additional references.

Case reports, case series, case–control studies, randomized trials, pharmacoepidemiologic studies, and systematic reviews were included. We did not use any language restriction in the search filter. Among the different drug categories, only papers involving antibiotics, antifungals, antiparasitics, and antivirals were selected. Results were subsequently divided by pharmacologic class according to mechanism of action, and for each drug for which there was at least one case report, an additional search was performed for the additional available evidence. The association with DIAP was defined for each drug using the classification criteria of the different classifications proposed in the literature. We did not include non-original reports or animal model studies. 

Titles and abstracts were independently assessed by three reviewers (A.D.G., C.C., and F.D.V.) to determine the eligibility of the studies. Both investigators checked the fulfillment of inclusion and exclusion criteria; in the case of doubt, the full text of the articles was retrieved and reviewed. A fourth author (E.C.N.) arbitrated in all the cases for a lack of agreement.

Data from eligible studies were independently extracted by three reviewers (A.D.G., C.C., and F.D.V.) and then were cross-checked. Discrepancies were rectified by consensus. If the article grouped patients from a previous study and newly enrolled ones, only the latter were considered. In the case of mixed cohorts, only data regarding patients who respected the eligibility criteria were included in the analysis.

## 7. Conclusions

There is little evidence for most drugs mentioned in the literature that are identified as triggers of AP. For antimicrobial and antiviral drugs, a total of 51 associations were reported. The drugs with the highest evidence are didanosine, protease inhibitors, and metronidazole. However, we highlight the additional drugs for which there is more evidence of a positive rechallenge in case reports including Isoniazid, tetracyclines, trimethoprim/sulfamethoxazole and erythromycin, lamivudine, pentamidine, stibogluconate, and voriconazole. For the remaining drugs, there is conflicting information because the assignment to a specific probability class in the articles reviewed is left to the judgement of the investigators on the quality of the case reports, and on different inclusion criteria. As a result, there is tremendous variability among current classifications, with the same drug being assigned to significantly different probability classes based on the same reports. Such problems arise with classification models based only on case reports, as in Wolfe et al. and Simons-Linares et al., and as it is well known that these types of papers represent the lowest level of scientific evidence. In addition, the number of DIAP reports for some drugs may also be increasing because of perceptions by the medical community. However, an arbitrary number of case reports is not necessary to increase the strength of evidence for an association and to distinguish it from a causal and nonrandom association. These are the reasons why randomized controlled trials are considered the gold standard. In their classification, Saini et al. created probability classes by considering not only case reports but also cohort studies, pharmacoepidemiologic analyses, and RCTs and thus differed from previous classifications. However, this approach also has its limitations, such as the fact that some studies are not designed to identify AP cases and that investigator reports may be underestimated. In conclusion, we believe that the results of multiple reports of AP cases during therapy should be considered as a basis for conducting prospective randomized controlled trials, improving the quality of evidence. In this way, helpful information can be obtained for the physician to decide whether the drug should be continued or possibly discontinued. In addition, for drugs with better evidence, most mechanisms of harm are still suspected or unknown. Further studies are needed to confirm the suspected etiopathogenetic mechanisms. 

## Figures and Tables

**Figure 1 antibiotics-12-01495-f001:**
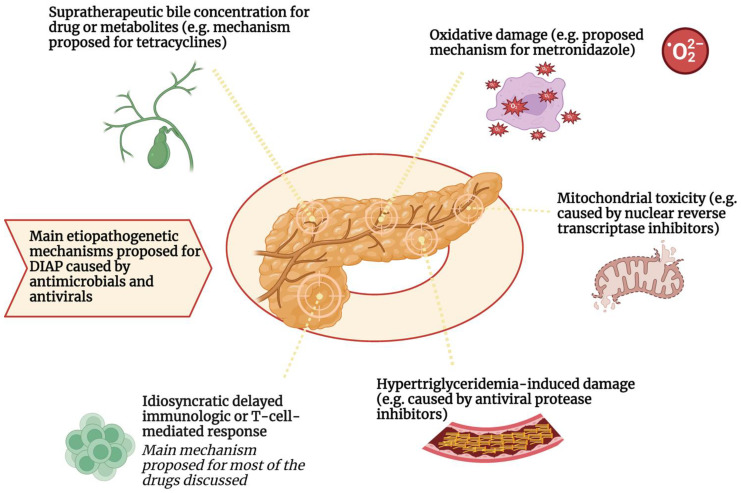
Main etiopathogenetic mechanisms proposed for DIAP caused by antimicrobials and antivirals.

**Table 1 antibiotics-12-01495-t001:** The classification of drug-induced pancreatitis, according to Badalov, updated by Simons-Linares et al. [13].

Class	Description
Ia	At least 1 case report with positive rechallenge, with exclusion of all other causes.
Ib	At least 1 case report with positive rechallenge, failing to document exclusion of other causes or other possible etiologies were available.
II	At least 4 cases in the literature with consistent latency.
III	At least 2 cases in the literature with no consistent latency among cases and no rechallenge.
IV	Drugs not fitting into the earlier described classes.

**Table 2 antibiotics-12-01495-t002:** The classification of drug-induced pancreatitis, according to Wolfe et al. [11].

Class	Description
Ia	At least 1 case report with positive rechallenge, with exclusion of all other causes.
Ib	At least 1 case report with positive rechallenge, failing to document exclusion of other causes or other possible etiologies were available.
Ic	At least 1 case report in humans, without a positive rechallenge, other causes are ruled out.
II	At least 2 cases in humans reported in the literature, without a positive rechallenge, with consistent latency, and other causes, were not ruled out.
III	At least 2 cases in humans reported in the literature, without a positive rechallenge, with inconsistent latency, and other causes, were not ruled out.
IV	Drugs not fitting into the earlier described classes.

**Table 3 antibiotics-12-01495-t003:** The classification of drug-induced pancreatitis, according to Saini et al. [9].

Class	Description
I	High quality of evidence for causation of acute pancreatitis: randomized controlled clinical trials.
II	Moderate quality of evidence for causation of acute pancreatitis: case–control studies and/or pharmacoepidemiology studies.
IIIa	Case reports showing “rechallenge and consistent latency”.
IIIb	Case report showing rechallenge only.
IIIc	Case report showing consistent latency only.
IV	Case Reports with no rechallenge or consistent latency.

**Table 4 antibiotics-12-01495-t004:** Antibiotics associated with acute pancreatitis.

Drug	At Least One Case with Positive Rechallenge	Studies Showing a High Probability of Association	Classification Class According to Simons-Linares et al.	Classification Class According to Wolfe et al.	Classification Class According to Saini et al.
**Antibiotics**					
Tetracicline	Yes	N/A	Not included	Ia	IIIb
Doxycicline	No	N/A	III	Ic	IIIc
Tigecycline	Yes	N/A	Ib	Ia	IV
Minocycline	No	N/A	III	Ic	Not included
Demeclocycline	No	N/A	Not included	IV	Not included
Erythromycin	Yes	N/A	III	Ia	IV
Clarithromycin	No	N/A	III	Ic	Not included
Roxithromycin	No	N/A	IV	IV	IV
Metronidazole	Yes	2 population-based case-control studies	Ia	Ia	II
Secnidazole	No	N/A	IV	Ic	IV
Tinidazole	No	N/A	IV	Ic	Not included
Ciprofloxacin	Yes	N/A	III	Ib	Not included
Levofloxacin	Yes	N/A	Not included	Not included	Not included
Ofloxacin	No	N/A	Not included	Not included	IV (with ornidazole)
Norfloxacin	No	N/A	Not included	IV	Not included
Nitrofuratoin	Yes	N/A	Ib	Ia	IIIb
Rifampicin	Yes	N/A	Not included	Ib	Not included
Trimethoprim/Sulfamethoxazole	Yes	N/A	Ia	Ia	IIIb
Amoxicillin-clavulanate	No	N/A	IV	Ic	Not included
Ceftriaxone	No	N/A	III	II	Not included
Ampicillin	Yes	N/A	III	Ib	Not included
Isoniazid	Yes	N/A			

N/A = not available.

**Table 5 antibiotics-12-01495-t005:** Antivirals associated with acute pancreatitis.

Drug	At Least One Case with Positive Rechallenge	Studies Showing a High Probability of Association	Classification Class According to Simons-Linares et al.	Classification Class According to Wolfe et al.	Classification Class According to Saini et al.
**Antivirals**					
Adefovir	No	N/A	IV	Ic	IV
Famciclovir	No	N/A	IV	IV	Not included
Lamivudine	No	N/A	IV	Ib	Not included
Didanosine	Yes	Yes	II	I	I
Remdesevir	No	N/A	Not included	Not included	Not included
Interferon α2b/ribavirin	No	N/A	III	Not included	IIIc
Ritonavir	No	Yes	III	IV	Not included
Indinavir	No	Yes	Not included	Not included	Not included
Nelfinavir	Yes	No	Ib	Ib	Not included
Telaprevir	Yes	No	Ia	Ia	Not included
Boceprevir	No	No	IV	Ic	Not included

N/A = not available.

**Table 6 antibiotics-12-01495-t006:** Antifungals and antiparasitics associated with acute pancreatitis.

Drug	At Least One Case with Positive Rechallenge	Studies Showing a High Probability of Association	Classification Class According to Simons-Linares et al.	Classification Class According to Wolfe et al.	Classification Class According to Saini et al.
**Antifungals and** **Antiparasitics**					
Pentamidine	Yes	N/A	Ib	Ib	IV
Meglumine antimoniate	Yes	N/A	II	Ib	Not included
Paromomycin	Yes	N/A	IV	Ib	Not included
Stibogluconate	Yes	N/A	Ib	Ib	Not included
Itraconazole	Yes	N/A	Not included	Ic	IV
Voriconazole	Yes	N/A	Not included	Ib	Not included
Artesunate	No	N/A	IV	Ic	Not included

N/A = not available.

## Data Availability

Not applicable.

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
