# Peer review of "Drug-Induced Acute Pancreatitis in Adults: Focus on Antimicrobial and Antiviral Drugs, a Narrative Review"

_antibiotics, 2023, doi:10.3390/antibiotics12101495_

Round 1

Reviewer 1 Report

Dear authors,

I was pleased to review the article „DRUG-INDUCED PANCREATITIS IN ADULTS: FOCUS ON ANTIMICROBIAL AND ANTIVIRAL DRUGS”

I read this article carefully and with great interest, the topic of acute pancreatites and drug-induced acute pancreatites is very important for the treatment of the patient with various diseases, both in the institution of different drugs and in the monitoring of adverse effects - in this case acute pancreatitis.

The methodology used by the authors is appropriate for the purpose of the study, with a very laborious information, with many details as well as many bibliographic references. 

I suggest some revisions:

-     -    the introduction of the term ACUTE  in the title of the article – ACUTE PANCREATITIS

-    -     for tables 1-3, isn't there a possibility to be merged-united?  through this, the differences between them can be highlighted more easily

-        - line 175-178 – „we have divided antibiotics, according to...” I suggest a bibliographic reference for this classification.

Author Response

Dear authors,

I was pleased to review the article „DRUG-INDUCED PANCREATITIS IN ADULTS: FOCUS ON ANTIMICROBIAL AND ANTIVIRAL DRUGS”

I read this article carefully and with great interest, the topic of acute pancreatites and drug-induced acute pancreatites is very important for the treatment of the patient with various diseases, both in the institution of different drugs and in the monitoring of adverse effects - in this case acute pancreatitis.

The methodology used by the authors is appropriate for the purpose of the study, with a very laborious information, with many details as well as many bibliographic references

I suggest some revisions:

-     -    the introduction of the term ACUTE  in the title of the article – ACUTE PANCREATITIS

Response: We thank the reviewer for the comment. We modified the title as suggested.

-    -     for tables 1-3, isn't there a possibility to be merged-united?  through this, the differences between them can be highlighted more easily

Response: We thank the reviewer for the precious suggestion. We created three further tables, where we entered all three classifications for each drug class

-        - line 175-178 – „we have divided antibiotics, according to...” I suggest a bibliographic reference for this classification.

Response: We apologize to the reviewer for this inaccuracy. We added the missing reference.  

Reviewer 2 Report

Manuscript title: DRUG-INDUCED PANCREATITIS IN ADULTS: FOCUS ON 2

ANTIMICROBIAL AND ANTIVIRAL DRUGS

The study is about the Acute pancreatitis (AP), an acute inflammation of the pancreas caused by the f antimicrobial and  antivirals drugs.  Causal associations between drugs and pancreatitis are largely based on case reports or case series with a prevalence of up to 5% and is of limited evidence. The available data was reviewed and presented. 51 suspected associations were observed between antimicrobial and antiviral drugs and AP. The drugs with the most evidence of correlation are: didanosine, protease inhibitors, and metronidazole. However, there are major differences between the various classifications available, where the same drug being assigned to different probability classes. It is likely that the presence of multiple case reports of an association between acute pancreatitis and a drug should serve as a basis for conducting prospective randomized controlled trials to improve the quality of the evidence.

With al the limitations with respect to the availability of the data, a fair review has been made and presented in a god manner without any ambiguity.  In my opinion the manuascript may be accepted for publication.  The  following suggestions has been made as value addition.

1.      Since only three drugs 6-mercaptopurine, azathioprine, and didanosine have been associated with the development of AP , the compounds structure could be given.

2.      It was mentioned that Most of the mechanisms underlying drug-induced pancreatic injury have not been fully  elucidated or demonstrated.  Atleast some more inputs  can be added into the manuscript with respect to the above three drugs.

3.      Attempts can be made to find out is there any structure activity relationship is  there.

Author Response

REVIEWER 2

The study is about the Acute pancreatitis (AP), an acute inflammation of the pancreas caused by the f antimicrobial and  antivirals drugs.  Causal associations between drugs and pancreatitis are largely based on case reports or case series with a prevalence of up to 5% and is of limited evidence. The available data was reviewed and presented. 51 suspected associations were observed between antimicrobial and antiviral drugs and AP. The drugs with the most evidence of correlation are: didanosine, protease inhibitors, and metronidazole. However, there are major differences between the various classifications available, where the same drug being assigned to different probability classes. It is likely that the presence of multiple case reports of an association between acute pancreatitis and a drug should serve as a basis for conducting prospective randomized controlled trials to improve the quality of the evidence.

With al the limitations with respect to the availability of the data, a fair review has been made and presented in a god manner without any ambiguity.  In my opinion the manuascript may be accepted for publication.  The  following suggestions has been made as value addition.

  1. Since only three drugs 6-mercaptopurine, azathioprine, and didanosine have been associated with the development of AP , the compounds structure could be given.

  1. It was mentioned that Most of the mechanisms underlying drug-induced pancreatic injury have not been fully  elucidated or demonstrated.  At least some more inputs can be added into the manuscript with respect to the above three drugs.
  2. Attempts can be made to find out is there any structure activity relationship is  there.

Response: Thanking the reviewer for the interesting suggestion, we have added further molecular and pharmacokinetic details for didanosine and a mention of the possible mechanism of didanosine-mediated pancreatic damage in the specific chapter. However, the literature is very scarce on this mechanism so we prefer not to add too many details.

We have not added further specifics on azathioprine and 6-mercaptopurine as t this review was focused on antivirals and antibiotics

Reviewer 3 Report

The authors have produced an excellent overview of pancreatic damage caused by anti-infectious drugs.

It's a very interesting study, and the subject is not very common in the literature.

A few revisions are worth adding:
- The title does not explain the nature of the study: the authors should mention whether it is a review , systematic , narrative ...

- It would be interesting to mention the keywords used by the authors to find cases and studies: is it an exhaustive study? search engines?

- Tables must add the author's reference.

- An explanatory figure on the mechanisms of action of anti-infectives is highly interesting to give greater clarity to the text.

- Is there a particular relationship between the severity of pancreatitis for each agent?

- Have there been any cases of post-viral covid or post-vaccine pancreatitis? It would be interesting to mention these cases, since the pandemic is still actual.

- It would be interesting to summarize the drugs in several tables for each type, giving the main results obtained.

- there are a few skipped periods and commas, so a minor revision of the text is suggested.

- Results and Conclusions are supported by the litterature. 

After minor changes the study could be accepted.

Needs minor revisions.

Author Response

The authors have produced an excellent overview of pancreatic damage caused by anti-infectious drugs.

It's a very interesting study, and the subject is not very common in the literature.

A few revisions are worth adding:

- The title does not explain the nature of the study: the authors should mention whether it is a review , systematic , narrative ...

Response: We thank the reviewer for pointing out this inaccuracy. The manuscript is a comprehensive and exhaustive narrative review, we added the term to the title to make it clearer.

- It would be interesting to mention the keywords used by the authors to find cases and studies: is it an exhaustive study? search engines?

Response: We thank the reviewer for the precious comment. Our manuscript is a comprehensive and exhaustive narrative review of the topic. We took the opportunity to detail the "materials and methods" section by writing in full detail all the keywords used in our search, as well as the search engines utilized.

- Tables must add the author's reference.

Response: We apologize to the reviewer for this inaccuracy. All the references have been added in the tables.

- An explanatory figure on the mechanisms of action of anti-infectives is highly interesting to give greater clarity to the text.

Response: Thanking the reviewer for the interesting purpose, we have added an explanatory picture regarding the main etiopathogenetic mechanisms postulated for DIAP caused by antimicrobials and antivirals.

- Is there a particular relationship between the severity of pancreatitis for each agent?

Response: We thank the reviewer for raising this interesting point. Unfortunately, for almost the totality of the drugs included in our manuscript, there is no relationship between the severity of pancreatitis and the molecule itself. Indeed, cases of both mild and severe acute pancreatitis have been described in the literature for all the drugs included in our review. Furthermore, the number of DIAP cases reported for each molecule is too small to make it possible to create an appropriate risk gradient of pancreatitis severity. Therefore, we believe that large animal studies on this topic could be crucial to achieve a better understanding of the mechanism of action underlying these otherwise unpredictable adverse drug reactions as well as to predict the severity of acute pancreatitis induced by each individual drug.

- Have there been any cases of post-viral covid or post-vaccine pancreatitis? It would be interesting to mention these cases, since the pandemic is still actual.

Response:  We thank the reviewer for the suggestion, we have added more epidemiological information on Sars Cov2 pancreatitis. We prefer not to add information on post-vaccine pancreatitis due to conflicting data in the literature

- It would be interesting to summarize the drugs in several tables for each type, giving the main results obtained.    

Response: We thank the reviewer for the interesting suggestion. We added three new table to the manuscript which further describes in greater detail the cases included in the study, dividing them according to the drug-category.

- there are a few skipped periods and commas, so a minor revision of the text is suggested.

Response: We apologize to the reviewer for these inaccuracies. We revised the whole manuscript as suggested.

- Results and Conclusions are supported by the litterature. 

After minor changes the study could be accepted.